# Analysis of Urban Residents' Travelling Characteristics and Hotspots Based on Taxi Trajectory Data

**Jiusheng Du \*, Chengyang Meng**  **and Xingwang Liu**

School of Surveying and Land Information Engineering, Henan Polytechnic University, Jiaozuo 454003, China; 212204010004@home.hpu.edu.cn (C.M.); 311705000230@home.hpu.edu.cn (X.L.)
\* Correspondence: djs@hpu.edu.cn

**Abstract:** This study utilizes taxi trajectory data to uncover urban residents' travel patterns, offering critical insights into the spatial and temporal dynamics of urban mobility. A fusion clustering algorithm is introduced, enhancing the clustering accuracy of trajectory data. This approach integrates the hierarchical density-based spatial clustering of applications with noise (HDBSCAN) algorithm, modified to incorporate time factors, with kernel density analysis. The fusion algorithm demonstrates a higher noise point detection rate (15.85%) compared with the DBSCAN algorithm alone (7.31%), thus significantly reducing noise impact in kernel density analysis. Spatial correlation analysis between hotspot areas and paths uncovers distinct travel behaviors: During morning and afternoon peak hours on weekdays, travel times (19–40 min) exceed those on weekends (16–35 min). Morning peak hours see higher taxi utilization in residential and transportation hubs, with schools and commercial and government areas as primary destinations. Conversely, afternoon peaks show a trend towards dining and entertainment zones from the abovementioned places. In the evening rush, residents enjoy a vibrant nightlife, and there are numerous locations for picking up and dropping off people. A chi-square test on weekday travel data yields a *p*-value of 0.023, indicating a significant correlation between the distribution of travel hotspots and paths.

**Keywords:** track data; spatiotemporal feature analysis; hotspot areas; hotspot paths

## 1. Introduction

The accelerated urbanization accompanying economic growth has significantly improved the quality of life in cities. However, this progress has also led to a substantial increase in urban population and vehicle numbers, contributing to severe traffic congestion and an imbalance in transportation resource allocation [1]. These challenges significantly affect residents' daily lives, incurring increased living costs and demanding advanced solutions for modern smart city development [2].

Understanding the spatiotemporal patterns, hotspot areas, and hotspot paths of urban travel is essential for effective urban transportation planning, public infrastructure management, and smart city development [3]. Current studies on urban residents' travel patterns primarily focus on three areas:

(1) Impact of environmental factors on commuting: This stream of research underscores how changing weather and seasonal variations influence commuter behavior. Feng et al. utilized Harbin's subway data to analyze the effect of adverse weather on public transportation, creating the WL long short-term memory model for forecasting commuter traffic during inclement weather conditions like rain and snow, thus assessing the impact on residents' preference for rail transit [4]. Similarly, Lin et al. conducted a study in the Greater Bay Area of Guangdong, Hong Kong, and Macau. They used the mixed geographically weighted regression (MGWR) model to investigate how various weather conditions, such as rain, snow, wind, and increasing temperatures, impact urban commuting patterns both immediately and over time. Their results

shed light on how these meteorological changes influence travel behaviors in different geographic areas [5].

(2) Behavioral pattern studies for travel forecasting: Focusing on the analysis and prediction of residents' travel characteristics, this type of research is crucial for enhancing public transit efficiency and offering customized travel services. Nithin et al. employed non-negative tensor decomposition (NTD) methods to visually represent the travel patterns of bus passengers in different areas of Davangere City over various time periods, including hours, days, and months. This approach made it easier to analyze how the mobility patterns of residents in the same areas changed across different seasons and allowed them to predict the spatial and temporal variations in resident movements in the near future [6]. Likewise, Ayad et al. leveraged advanced deep recurrent neural network techniques, which were trained on extensive resident travel data, to effectively capture the evolving and sequential characteristics of urban traffic flow. This effort resulted in the creation of a highly accurate urban traffic model, achieving an impressive accuracy rate of up to 95% [7].

(3) Trajectory data mining and personalized travel data collection: Traditionally, when mining trajectory data for transportation analysis, researchers relied on data from questionnaires and fixed-route buses, which failed to efficiently capture personalized travel information of residents. However, with the rapid advancement of network technology and the public transportation industry in recent years, it has become more convenient to collect personalized travel data from sources such as taxi global navigation satellite system (GNSS) trajectory data, residents' mobile signaling data, and social media data [8,9]. In one study, Zhu et al. used shared bicycle travel data of residents and introduced the time series weighted regression (TSWR) model to address the challenge of sparse statistical data when making long-term predictions. They enhanced the model's accuracy by over 35% by incorporating the rule-based adjustment optimization (RAO) method to refine nonlinear components, considering various factors, compared with using the RAO method alone [10]. In another study, Hu et al. developed the position opportunity selection (POS) model by taking into account both the population and quantity of points of interest (POI). They obtained residents' travel data in Harbin through the Unicom Smart Footprint platform. By comparing the POS model's analysis and predictions of residents' travel activities with actual behaviors, they found that the POS model's results were generally consistent with real-world observations [11].

Taxis have increasingly become a preferred mode of transport for many due to their immediate availability, ease of use, and adaptable scheduling. Their 24/7 availability and direct point-to-point service further enhance their appeal [12,13]. Consequently, analyzing taxi trajectory data to understand urban residents' travel patterns has emerged as a significant research method [14–16]. Residents' travel behavior covers both temporal aspects, reflecting travel duration and frequency, and spatial aspects, indicating movement patterns and dynamic trajectories (manifested as hotspot areas and paths). These dual characteristics unveil the intricate travel patterns of residents and identify urban travel hotspots [17]. Traditional methods for analyzing travel characteristics and hotspots from trajectory data involve data normalization using taxi GPS data, extraction of travel features, and clustering of trajectory points using algorithms like K-means or DBSCAN. In the end, the clustering results are examined to identify patterns in the behavior of residents [18]. However, these methods face challenges such as dependence on data quality, which directly impacts analysis accuracy [19]; low clustering accuracy due to simplistic data clustering [20]; and limited processing capabilities when dealing with large-scale, high-dimensional data [21–23].

This paper delves into the utilization of taxi trajectory data to address issues related to the quality of trajectory data and clustering accuracy. It introduces a novel density clustering algorithm that integrates the hierarchical density-based spatial clustering of applications with noise (HDBSCAN) clustering algorithm and kernel density analysis. This integration effectively identifies hotspot areas for passenger pick-up and drop-off.

The study also explores the connections between these hotspot areas. The algorithm's methodology comprises three stages: First, it incorporates time threshold limitations into HDBSCAN to form clusters of trajectory points across various time periods and levels. Second, it employs kernel density analysis to conduct a detailed examination of these clusters. Finally, it visualizes the processed trajectory points on a heat map, facilitating a deeper analysis of residents' travel patterns. After verification using experimental data, the fusion algorithm achieved the following results:

(1) The HDBSCAN algorithm, with high efficiency in processing noise points, accurately identifies trajectory point clusters and noise points in taxi trajectory data on both weekdays and weekends. It generates three distinct peak periods in the morning, middle of the day, and evening based on different time intervals for trajectory clustering.

(2) Building upon the clustering results, it eliminates noise points outside the trajectory data clusters and conducts kernel density analysis. This leads to the creation of a more accurate heat map compared with single kernel density analysis. It also identifies hotspot areas where residents frequently travel.

(3) Using the fusion algorithm, hidden Markov model road matching and hotspot detection were performed to determine hotspot paths. The chi-square test was employed to calculate the *p*-value for residents' travel hotspot areas and hotspot paths during various peak hours on weekdays, resulting in a value of 0.023, which is less than 0.05. This demonstrates a significant correlation between the two.

The paper is structured into several sections: Section 2 details the experimental setup and methodologies, Section 3 analyzes residents' travel time patterns and explores travel hotspots and paths, Section 4 discusses related research, and Section 5 concludes with a summary of the experimental process and findings.

## 2. Materials and Methods

The study outlines a four-stage process for analyzing residents' travel patterns and hotspots using taxi trajectory data, as depicted in Figure 1. The first stage focuses on data preprocessing, including the removal of default values, duplicates, redundant values, noise, and outliers. The second stage conducts a time feature analysis, segmenting data into larger and smaller time frames to extract origin–destination (OD) points and identify residents' travel time patterns. The third stage involves density-based cluster analysis (using the HDBSCAN algorithm) and kernel density analysis (employing a Gaussian-function-based method) on selected taxi OD points to determine areas of high travel activity. The final stage delves deeper into these hotspots, performing tasks like road network matching, hotspot detection, and visualization to identify hotspot paths.

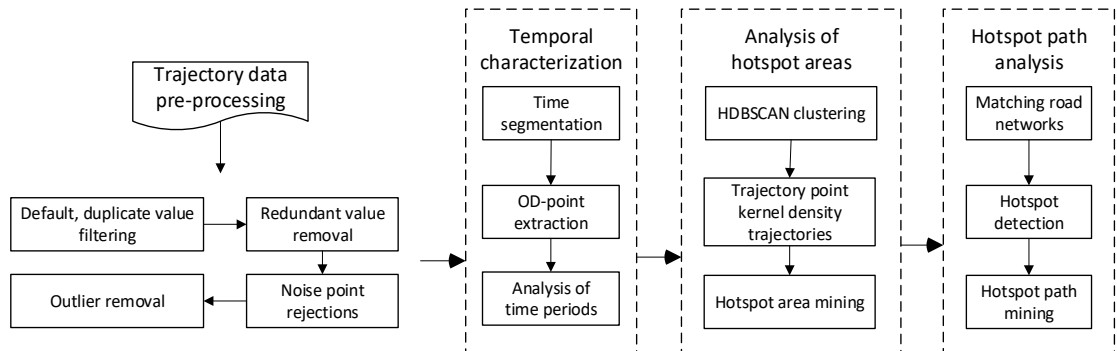

**Figure 1.** Technology roadmap.

### 2.1. Preprocessing of Trajectory Data

The original taxi trajectory data suffer from inaccuracies and poor quality due to the presence of default values, duplicates, redundant data, noise points, and outliers that

deviate from the usual trajectory sequence. If not addressed promptly, these issues can significantly impact subsequent research [24]. The preprocessing involves the following:

(1) Deletion of default and duplicate values: Track data are examined segmentally to remove rows with missing attributes and duplicate records. Table 1 illustrates data defaults (first row) and typical duplicate records (second and third rows). Furthermore, the absence of data in any column can be referred to as a default value.

(2) Removal of redundant values: Trajectory data often accumulate redundant points when taxis stop at gas stations, shops, or traffic lights or in traffic jams. To streamline data processing, a novel speed attribute was introduced for calculating taxi speeds using timestamp intervals. A speed threshold was established to eliminate redundant data, setting minimum and maximum speed limits at 6 and 16.7 m/s, respectively. These limits were chosen based on the speed restrictions of various roads in the study area. Any data not falling within this speed range will be discarded.

(3) Elimination of noise points: Noise points, typically arising from taxi collection terminal signal anomalies, equipment failures, or unusual driver behavior, deviate noticeably from the normal trajectory pattern. These points are identified and removed [25], as exemplified in Figure 2.

(4) Deletion of outliers: Data representing abnormal taxi trajectories, such as unchanged passenger status for an extended period, distances of over 2 km between adjacent points, or frequent, irregular changes in taxi passenger status over a short duration, are identified and removed [26].

**Table 1.** Default and duplicate data.

| cab_id | Latitude | Longitude | Status | Timestamp |
|---|---|---|---|---|
| 5912 | 30.631813 | 104.030495 | Nun | 3 August 2014 21:51:07 |
| 6313 | 30.631852 | 104.035066 | 1 | 3 August 2014 22:58:45 |
| 6313 | 30.631852 | 104.035066 | 1 | 3 August 2014 22:58:45 |

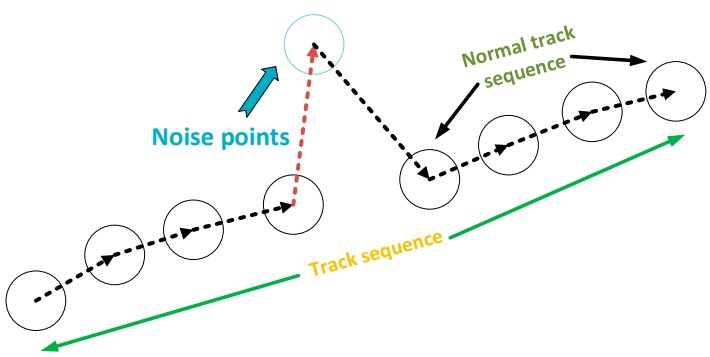

**Figure 2.** Schematic diagram of noise points.

## 2.2. Extraction and Visualization of Passenger Pick-Up and Drop-Off Points

The status field in the taxi trajectory data indicates the passenger status of the taxi, changing when a passenger boards or disembarks. These status changes mark the OD of passengers' trips, with the intermediate data points representing the specific travel path. An unoccupied taxi is denoted by a status of 0, changing to 1 when carrying passengers. When a taxi with the same cab_id finishes its passenger-carrying activity, the status will transition from 0 to 1 and then back to 0. The passenger pick-up and drop-off process, illustrated in Figure 3, includes the following:

(1) The taxi begins its operation with an initial status of 0.

(2) The taxi picks up passenger A, changing its status to 1, marking the starting point O1.

(3) After traveling a distance, passenger A disembarks, and the status reverts to 0, indicating the endpoint D1.

(4)  After driving for a certain distance, the taxi then picks up passenger B, changing the status back to 1, marking another starting point O2.
(5)  Upon reaching the destination, passenger B disembarks, and the status returns to 0, marking the endpoint D2.
(6)  This process repeats throughout the taxi's operational day.

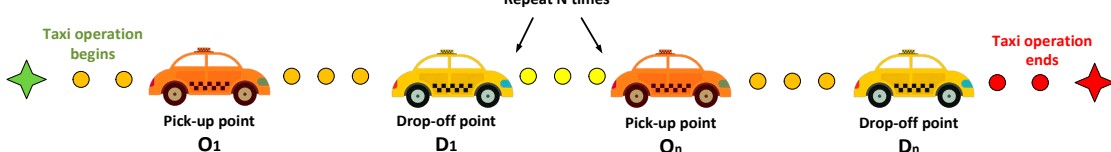

**Figure 3.** Taxi operation diagram.

All OD points together form the complete trajectory data for the taxi, which accurately records the travel information of all passengers in the city over a day. These OD points in the trajectory data are programmatically separated, stored, and visualized, as depicted in Figure 4.

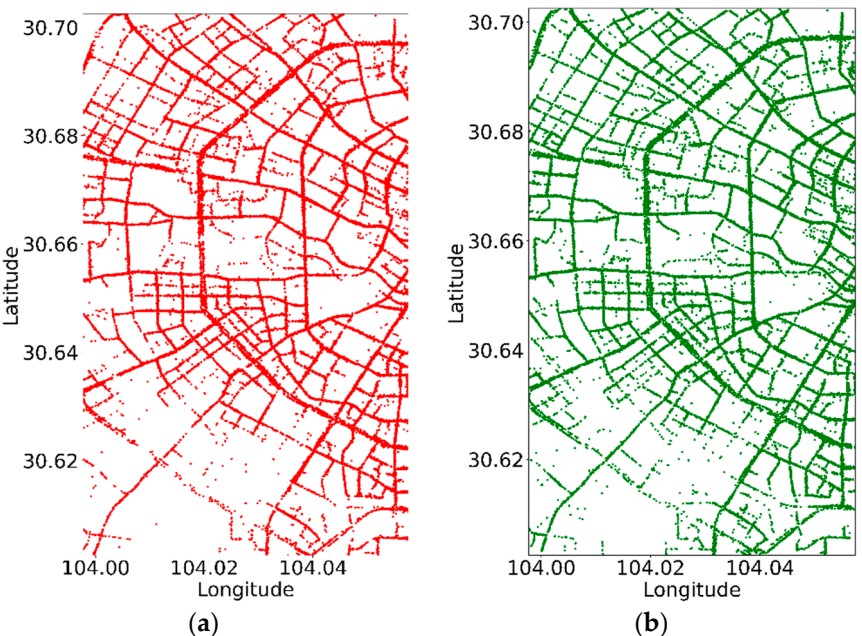

**Figure 4.** OD point visualization: (**a**) visualization of picking-up points; (**b**) visualization of drop-off points.

### 2.3. Road Network Matching

Road network matching is crucial for analyzing and understanding travel behaviors. By aligning trajectory data with the road network, we can accurately determine the start and end points of residents' travels, along with their paths, providing insights into travel characteristics, hotspot paths, and pick-up/drop-off locations. The trajectory data collected by the taxi terminal ideally correspond to the associated road network. However, due to positioning inaccuracies and signal interferences, some trajectory points may deviate from the actual roads. To counter this, the hidden Markov model is employed for road network matching [27]. The process involves the following:

(1)  Calculating transition probabilities: This step involves computing the transition probabilities between taxi trajectory points and multiple candidate paths on the map:

$$Pr\left(rss_{t,j}\middle|path_{t-1,i}\right) = e^{-\left|Dist(prj_t,prj_j)-Dist(P_{last},\,P_{first})\right|}, \tag{1}$$

where $P_{first}$ represents the first track point record, $P_{last}$ represents the last track point record, $prj_t$ represents the projection point of trajectory data on the map path, $Dist(prj_t, prj_j)$ represents the shortest distance between projection points $prj_t$ and $prj_j$ on the road network, and $Dist(P_{last}, P_{first})$ represents the Euclidean distance between the starting trajectory point and the ending trajectory point of a certain path.

(2) Observation probability calculation: Here, the observation probability between the taxi trajectory points and multiple corresponding paths is determined:

$$Pr(traj_t | rss_{t,j}) = \frac{1}{\sqrt{2\pi}\sigma_z} e^{-\frac{1}{2}\left(\frac{Dist(traj_t, rss_{t,j})}{\sigma_z}\right)^2}, \tag{2}$$

where $Dist(traj_t, rss_{t,j})$ is the distance between the trajectory point and the corresponding path, and $\sigma_z$ is the standard deviation of the distance between the trajectory point and the corresponding path.

(3) Maximum probability matching: This step calculates the maximum probability for aligning the taxi trajectory points with the map paths:

$$\prod_{t=1}^{m} Pr(traj_t | path_{t,j}) * \prod_{t=2}^{m} Pr(path_{t,j} | path_{t-1,i}), \tag{3}$$

where $traj_t$ represents various trajectory segments, $path_t$ represents the candidate path, $Pr(traj_t | path_{t,j})$ represents the observation probability between the taxi trajectory and the map path, and $Pr(path_{t,j} | path_{t-1,i})$ represents the transition probability between adjacent taxi trajectory points and map paths. The concept underlying this formula is illustrated in Figure 5.

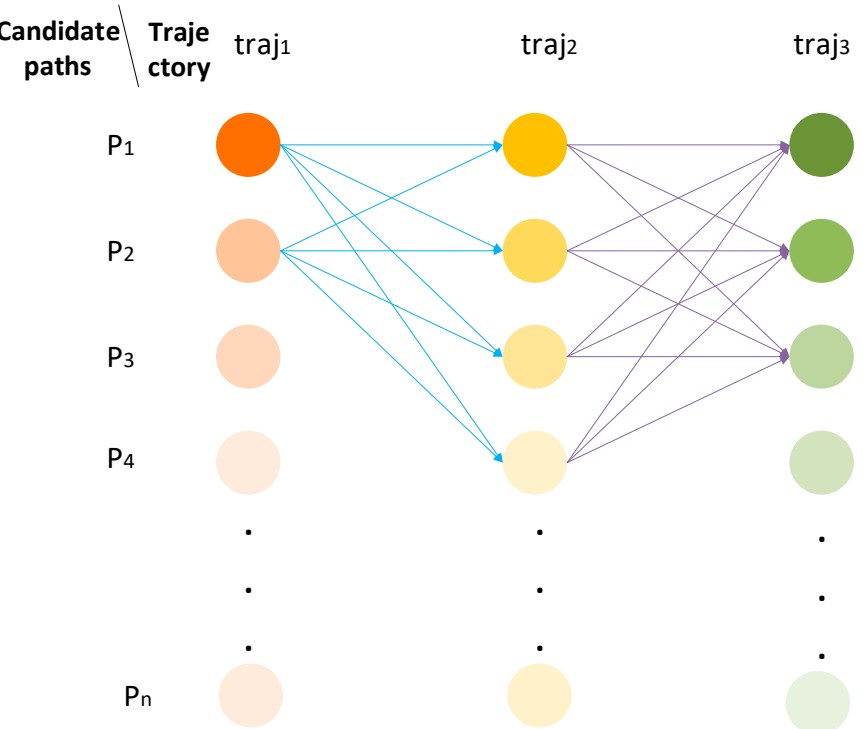

**Figure 5.** Schematic diagram of trajectory matching.

### 2.4. HDBSCAN Algorithm Cluster Analysis

The HDBSCAN algorithm, an advancement of the DBSCAN algorithm, is adept at clustering in areas with varying densities. The algorithm's operation includes the following:

(1) Matrix construction based on point distance.

Utilizing the spherical cosine law, the algorithm calculates the distance $D_{i,j}$ between two trajectory points with coordinates *(lat$_i$, lon$_i$)* and *(lat$_j$, lon$_j$)*; *R* represents the radius of the earth, which is about 6371 km.

$$D_{i,j} = cos^{-1}(sin(lat_1) \times sin(lat_2) + cos(lat_1) \times cos(lat_2)$$

$$\times \cos(lon_2 - lon_1)) \times R \tag{4}$$

This distance is then used to populate a symmetric matrix. Diagonal elements $D_{i,j}$ are set to zero, representing the distance from a point to itself.

(2)  Creation of a distance-weighted graph and minimum spanning tree (MST).

Trajectory points are interconnected based on the distance matrix to form a distance-weighted graph. Starting from an appropriate edge, the algorithm finds the edge with the minimum weight that connects the tree to vertices not yet in the tree. This edge is then added to the tree, and the process is repeated until all vertices are included. This process is illustrated in Figure 6.

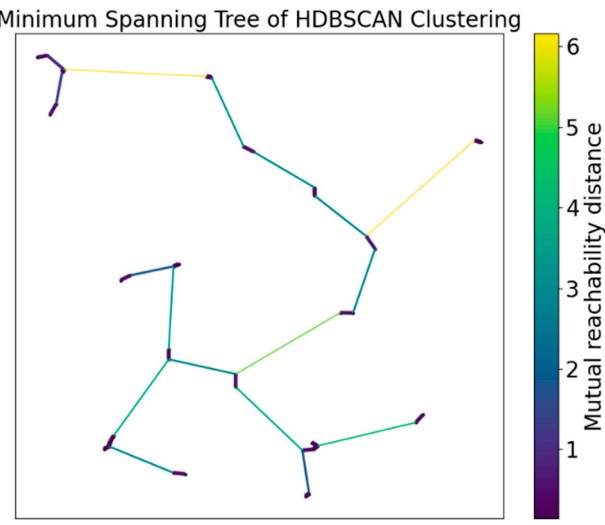

**Figure 6.** Minimum tree diagram.

(3)  Creation of an interconnectivity diagram.

This step involves defining the minimum neighbor density threshold (MinPts) based on core distances. A dense area is formed when the number of neighbors in a cluster exceeds MinPts. The reachability distance is defined as the maximum of the core distance and the actual distance between two endpoints in the MST. If both endpoints are in high-density areas, the actual Euclidean distance is used; if one endpoint is in a low-density area, the larger core distance is chosen. The MST is then transformed into an interconnectivity graph based on this reachability distance.

(4)  Construction of hierarchical clustering tree.

Utilizing the interconnectivity graph, a hierarchical clustering tree is constructed, as presented in Figure 7. This tree represents the connections of data points at various levels, facilitating a comprehensive understanding of the clustering process from individual points to the entire database.

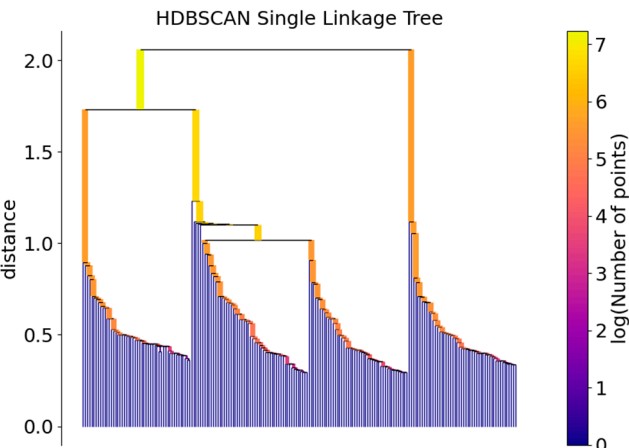

**Figure 7.** Hierarchical clustering tree connection diagram.

(5)     Simplification of the clustering tree hierarchy.

The stability and duration of each cluster in the hierarchical tree are analyzed. Clusters with short durations or instability are eliminated, focusing on more distinct and stable data clusters.

(6)     Optimal clustering selection.

The final clustering results are chosen based on cluster stability, ensuring that the selected clusters are of high quality and reliability.

To demonstrate the effectiveness, part of the trajectory data are subjected to both DBSCAN and HDBSCAN clustering for comparison (Figure 8).

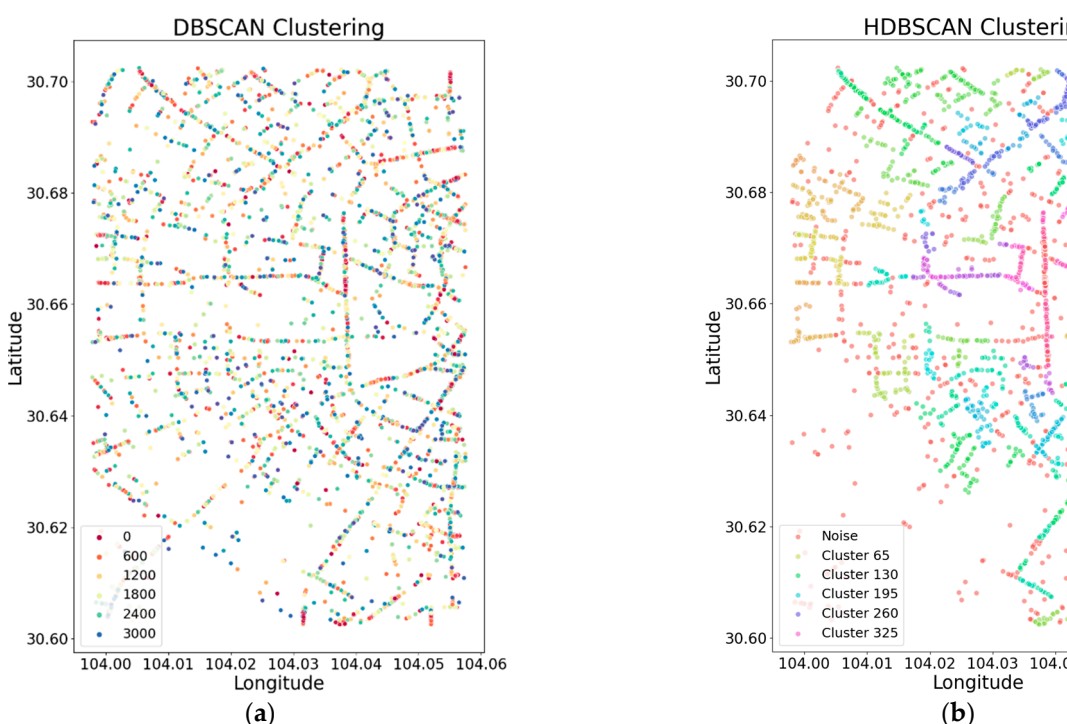

**Figure 8.** Comparison of different aggregation results: (**a**) DBSCAN clustering; (**b**) HDBSCAN clustering.

The two clustering methods yield noise point detection proportions of 15.85% and 7.31%, respectively. This comparison highlights the superior noise reduction capability of

the HDBSCAN algorithm. In Figure 8, we can observe that HDBSCAN clustering exhibits a higher point density, resulting in a more continuous clustering trajectory. This characteristic is advantageous for identifying residents' travel hotspots. On the other hand, DBSCAN clustering displays lower point density, causing the clusters to be more dispersed and making it challenging to visually identify high-density areas of trajectory points.

### 2.5. Kernel Density Analysis of Trajectory Points

Kernel density estimation (KDE) is a suitable technique for analyzing and visualizing geographical information data that include location information, such as trajectory data. It is a nonparametric method used to estimate the probability density function [28]. Kernel density analysis is valuable for identifying areas with varying densities, analyzing the frequency of different data points, and estimating the spatial density of point features within a series of trajectory points. In this paper, the kernel density analysis was conducted in conjunction with HDBSCAN clustering. Here are the main concepts introduced:

(1) Kernel function: This function assigns weights to each data point, creating a smooth distribution around them and estimating a continuous probability density function from discrete data. In this experiment, the Gaussian kernel, resembling a normal distribution curve and suitable for most scenarios, was used. The mathematical expression of the Gaussian kernel is shown in Equation (5):

$$K(u) = \frac{1}{\sqrt{2\pi}} e^{-\frac{1}{2}u^2},\qquad(5)$$

where $K(u)$ is the value of the kernel function at point $u$; $e$ is the base of the natural logarithm, approximately equal to 2.71828; $\pi$ is the pi ratio, approximately equal to 3.14159; and $u$ is the standardized distance from the origin, calculated as $\frac{x-x_i}{h}$, where $x$ is the evaluation point, $x_i$ is the data point, and $h$ is the bandwidth.

(2) Bandwidth selection: Bandwidth is a crucial parameter in KDE as it dictates the smoothness of the kernel function. An overly large bandwidth can result in overfitting, where the model excessively conforms to the data set. Conversely, a small bandwidth can lead to underfitting, failing to capture the underlying trends in the data. Therefore, choosing an appropriate bandwidth value is essential for accurate KDE.

(3) KDE calculation formula: The mathematical formulation of KDE is outlined in Equation (6):

$$f(x) = \frac{1}{nh}\sum_{i=1}^{n} K\left(\frac{x-x_i}{h}\right),\qquad(6)$$

where $f(x)$ represents the estimated density of trajectory points at position $x$, $n$ is the number of sample points, $x_i$ is the sample trajectory point, $h$ is the broadband parameter, and $K$ is the kernel function.

To demonstrate the effectiveness of this approach, 44,753 data points from weekdays were selected for a clustering comparison. Both single kernel density analysis and kernel density analysis after clustering (with noise points excluded) were performed. After noise point exclusion, 37,460 data points remained. In both clustering scenarios, the chosen bandwidth value h was 0.2, with all other conditions kept constant. The resulting kernel density estimations are presented in Figure 9.

By comparing Figure 9a,b, it becomes evident that after removing the noise points, the kernel density analysis provides a richer and more accurate representation of data point density. This is evident in the varying color depths, which more precisely reflect the density changes in the data points.

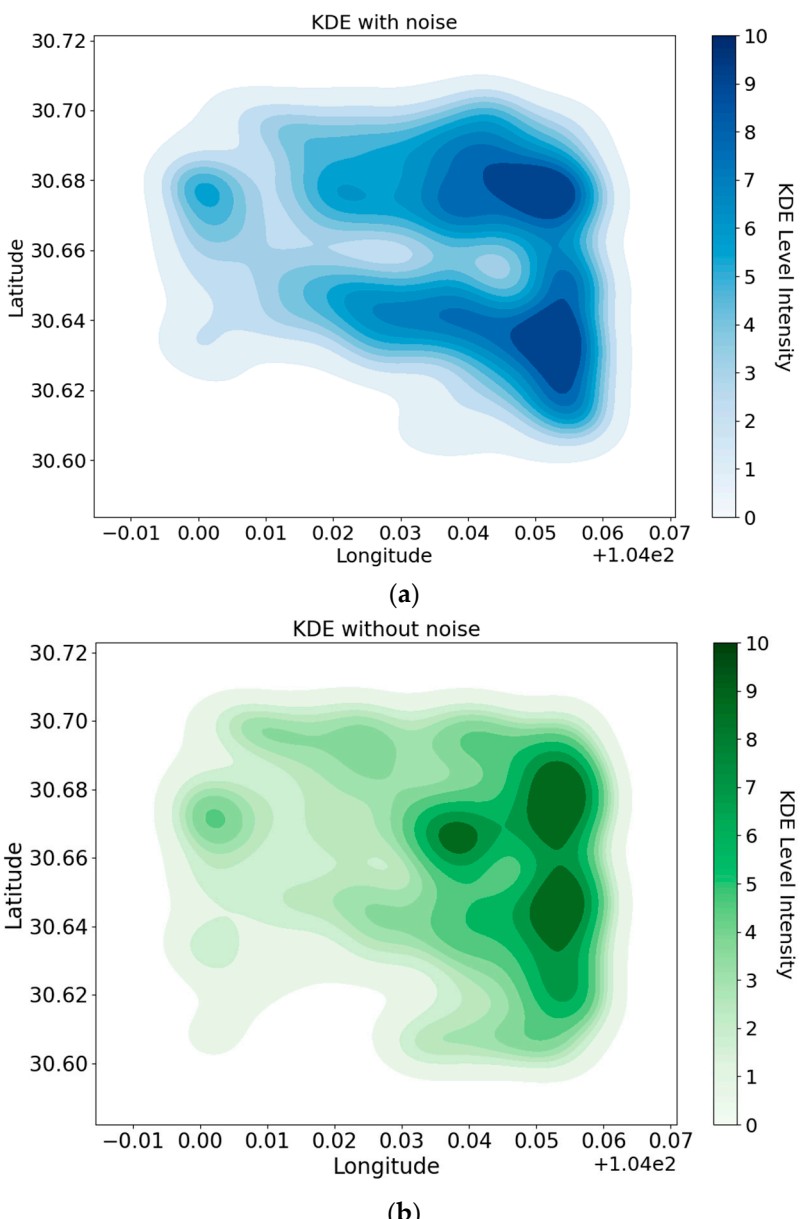

**Figure 9.** Comparison of before and after clustering and density analysis: (**a**) kernel density analysis before clustering; (**b**) kernel density analysis after clustering.

### 2.6. Chi-Square Test

The chi-square test is a statistical method employed to determine if there is a significant correlation between two categorical variables. It compares the observed frequencies with expected frequencies to assess if the observed data align with a theoretical model. This study used the chi-square test to analyze the distribution of hotspot areas and paths across different time periods. The chi-square test statistic and the formula for calculating degrees of freedom are as follows:

$$\chi^2 = \sum \frac{(O_i - E_i)^2}{E_i}, \tag{7}$$

$$df = (r - 1) \times (c - 1), \tag{8}$$

where $\chi^2$ represents the chi-square statistic that measures the difference between observed and expected frequencies, $O_i$ represents the observed frequency, and $E_i$ represents the expected frequency. The frequency in each category is calculated based on the theoretical model under the null hypothesis. In Equation (8), $r$ represents the number of rows and $c$

the number of columns, which are utilized to determine the critical value in the chi-square distribution table. After calculating the chi-square statistic and degree of freedom, Python software (version 3.7.0) was used to compute the *p*-value. If this *p*-value is less than the significance level (commonly set as 0.05), it suggests sufficient evidence to reject the null hypothesis and consider the two variables as related.

## 3. Experiments, Results, and Analysis

### 3.1. Data Sources and Structure

The experimental study focused on data collected within specific geographic coordinates in Jinniu District, Chengdu City, spanning latitude 30.60–30.70 and longitude 104.00–104.06. The temporal scope included 4 days, from 7 to 10 August 2014, with 7–8 August classified as weekdays and 9–10 August as weekends. The data comprised 15,190,410 entries gathered from approximately 7000 taxis. The data's attribute structure is detailed in Table 2.

**Table 2.** Data's attribute structure.

| Attribute Fields | Data Example | Data Types | Data Description |
|---|---|---|---|
| cab_id | 3949 | Long Int | Taxi collection terminal number |
| latitude | 30.628614 | Float | Latitude of data collected (GCJ-02 coordinate system) |
| longitude | 104.02519 | Float | Longitude of data collected (GCJ-02 coordinate system) |
| status | 0 | Int | Passenger loading status (0 means empty; 1 means loaded) |
| timestamp | 3 August 2014 22:50:54 | Timestamp | Timestamp of recorded data |

From Table 2, we can gather information about the longitude and latitude positions and the passenger status of a taxi collection terminal at a specific moment. However, it is important to note that the longitude and latitude coordinates in the trajectory data are in an encrypted format known as the GCJ-02 coordinate system. To perform map matching and subsequent analysis, we need to convert these coordinates to the WGS-84 coordinate system. Additionally, we can determine the sampling interval of the terminal data by analyzing the timestamp field in the taxi trajectory data. It appears that data with a 10 s interval makes up approximately 78% of the total data, constituting the majority. As a result, it is essential to filter out data with sampling intervals that are either too short or too long in accordance with the experimental requirements.

### 3.2. Analysis of Residents' Travel Time Characteristics

To investigate the travel characteristics of residents, we began by focusing on the temporal aspect. Utilizing the OD points derived from trajectory data, our initial analysis involved tracking variations in the number of taxi trips taken by residents across different dates and time intervals. Alterations in travel duration, taxi operation duration, and passenger load capacity were also examined. Additionally, we compared the proportion of passenger travel time and similar factors across various dates and time segments. This approach aims to provide a more comprehensive understanding of residents' travel patterns. Specifically, different types of days were considered, including both weekdays and weekends, and time periods were broken down into hourly intervals.

#### 3.2.1. Weekday Travel Demand Analysis

The study started by evaluating the total number of taxi operations on weekdays. Using the processed trajectory data, the total numbers of taxi trips were found to be 6597 and 6846 on the two weekdays, with passenger counts of 56,096 and 58,462, respectively. Next, the analysis focused on hourly travel patterns during weekdays, excluding the 24:00 to 6:00 night period due to low passenger numbers. The analysis included calculating the number of taxi trajectories per day, enabling the determination of passenger pick-up and drop-off trends across different times. These statistics are represented in Figure 10, providing visual insights into the fluctuating demand for taxi services throughout the day.

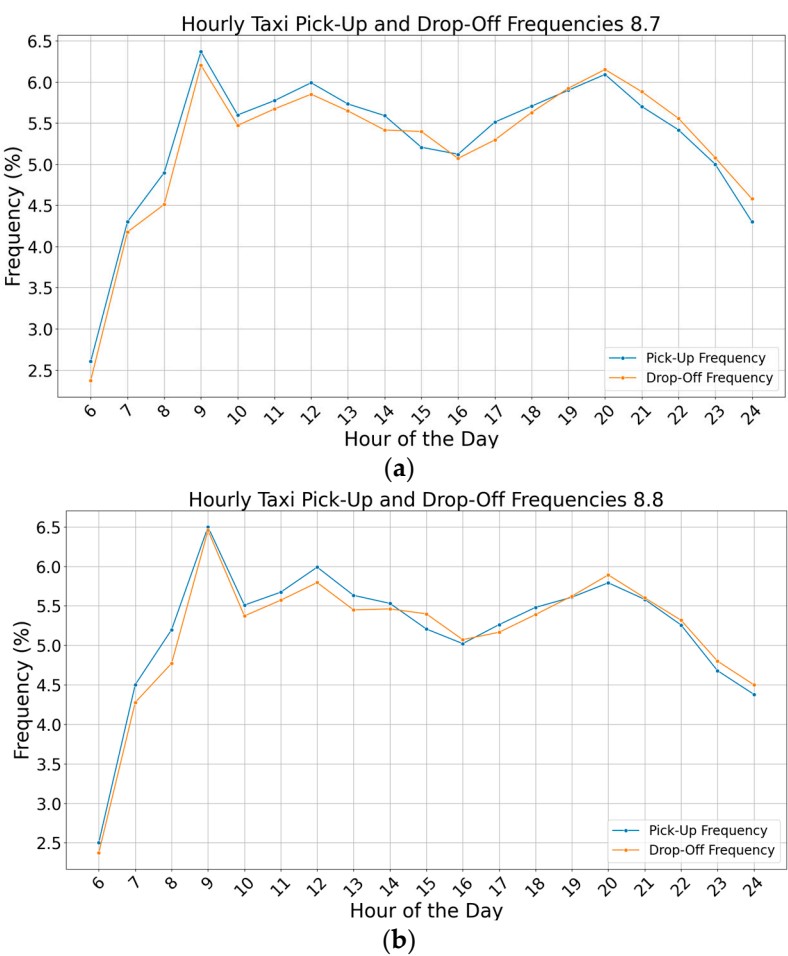

**Figure 10.** Statistical chart of passenger pick-up and drop-off on weekdays: (**a**) statistics of passenger pick-up and drop-off on 7 August; (**b**) statistics of passenger pick-up and drop-off on 8 August.

According to the data in Figure 10, a clear pattern in the frequency of taxi pick-up and drop-off points during different times on weekdays is observed, with noticeable peaks at around 7:00–9:00, 11:00–13:00, and 19:00–21:00. The specific analyses for these time periods are as follows:

(1) Travel volume experiences a marked surge after 6:00, reaching its zenith at around 9:00. This surge is attributed to residents commuting to work or school, creating a pronounced morning peak in taxi usage.

(2) There is a minor increase in the number of taxi passengers after 11:00, reaching its highest point at around noon, and then gradually decreasing after 13:00. This pattern corresponds to lunch breaks at workplaces and schools, leading to increased taxi usage as residents travel to dine and rest.

(3) There is a notable decrease in taxi passenger numbers during the evening, particularly between 14:00 and 17:00. This reduction is likely due to evening rush hour congestion, impacting the efficiency of taxi services.

(4) Another peak in taxi usage is observed between 19:00 and 21:00, coinciding with residents engaging in social and entertainment activities after work or school.

(5) Taxi passenger volume gradually decreases after 23:00, aligning with residents' typical bedtime routines on weekdays.

3.2.2. Weekend Travel Demand Analysis

The travel patterns of residents on weekends were analyzed by the divided time periods. Initially, it was determined that the total numbers of taxi trips on two weekends

were 4871 and 4739, respectively. Additionally, the total numbers of passenger rides were 39,065 and 36,197 for these respective days. We then conducted a detailed statistical analysis of pick-up and drop-off frequencies during specific time intervals, as illustrated in Figure 11:

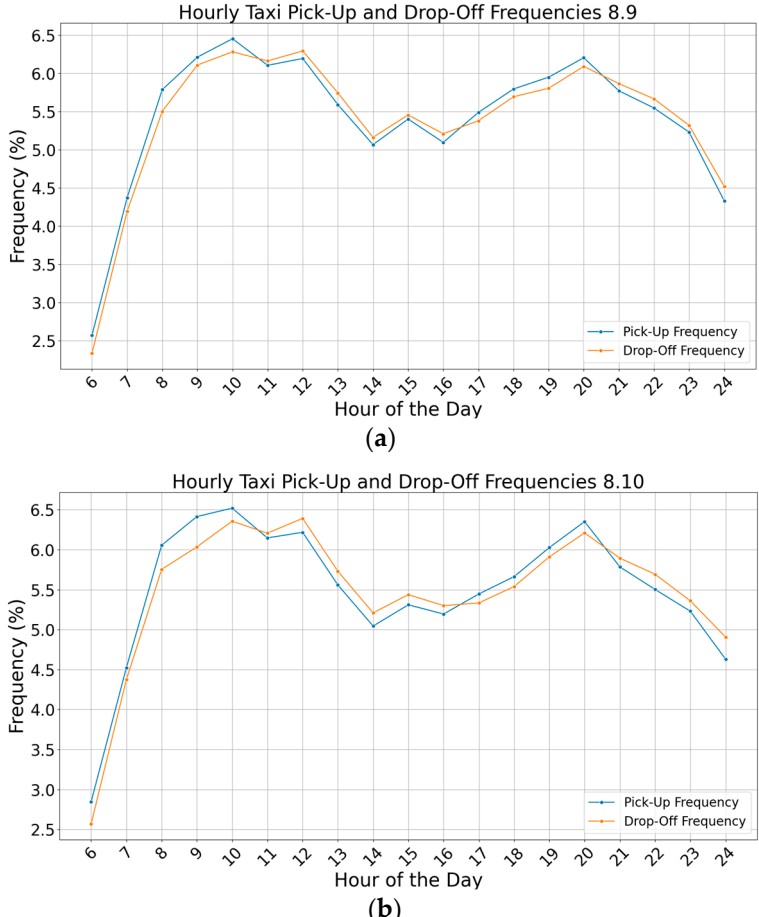

**Figure 11.** Statistics of passenger pick-up and drop-off on weekends: (**a**) statistics of passenger pick-up and drop-off on 9 August; (**b**) statistics of passenger pick-up and drop-off on 10 August.

It is evident from Figure 11 that the peak of fluctuation in weekend pick-up and drop-off can be observed during three distinct time intervals: 09:00–11:00, 14:00–16:00, and 19:00–21:00. To delve into further detail, the analysis is as follows:

(1)　On weekends, taxi usage increases gradually after 6:00, with a slower ascent around 9:00 and reaching a morning peak at 10:00. The delayed start is attributed to the additional rest time available to residents on weekends.

(2)　After 10:00, there is a slight decline in passenger volume, followed by a rebound after 11:00, peaking again at noon. This pattern reflects residents' leisure activities, such as visiting entertainment spots and dining venues and shopping during their off days.

(3)　A subdued peak forms at around 14:00, followed by another gradual increase at around 15:00, as residents opt to rest at home or continue their activities in entertainment or shopping venues.

(4)　A sharp rise in taxi usage is noted at around 19:00, with the number of passengers reaching an evening peak at 20:00. This increase is linked to residents visiting entertainment venues or dining out, followed by a return home after a day of leisure activities. These travel patterns on weekends align well with the typical leisure and rest behaviors of residents.

### 3.2.3. Comparative Analysis

To gain a comprehensive understanding of residents' travel patterns, a comparative analysis of taxi travel duration on both weekdays and weekends is essential. This involved calculating the passenger carrying time of taxis $i$ at different times of the day, denoted as $T_1, T_2, T_3, T_4, \ldots T_i$, with the cumulative duration for an entire day recorded as $\sum_{i=1}^{n} T_i = 1$. Using the OD point matching algorithm, the analysis focused on the statistical travel time of residents on taxis on both weekdays and weekends, as exemplified in Figure 12.

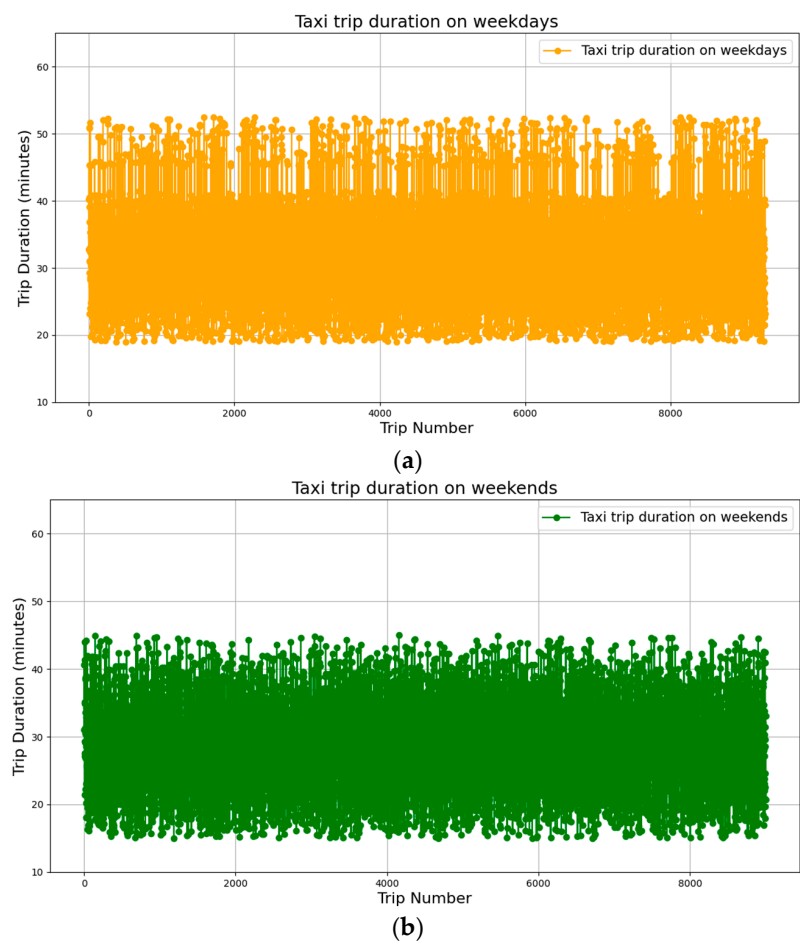

(**a**)

(**b**)

**Figure 12.** Statistics of residents' travel time on taxis: (**a**) statistics of residents' travel time on weekdays; (**b**) statistics of residents' travel time on weekends.

By comparing the passenger flow during various time periods on both weekdays and weekends, the following conclusions are obtained:

(1) On weekdays, peak hours are predominantly in the time periods of 7:00–9:00, 11:00–13:00, and 19:00–21:00. The uniformity in residents' travel purposes on these days, primarily for work or school, results in more regular and concentrated traffic flow. Time-bound obligations, such as office check-ins and school start times, contribute to this pattern.

(2) Weekends exhibit more varied peaks and troughs in passenger volume, and the timing of taxi usage tends to be later compared with weekdays. This variation stems from the lack of uniform travel purposes on weekends, giving residents the freedom to choose their destinations and travel times without strict time constraints. Consequently, weekend travel is characterized by more rest time and flexibility.

(3) Combining the findings from Sections 3.2.1 and 3.2.2 with Figure 12, it is observed that weekday travel times predominantly range from 19 to 40 min, while weekend travel times are mostly between 16 and 35 min. The longer travel times on weekdays could be attributed to longer commutes to work or school. In contrast, weekend

travels are typically shorter city trips. Additionally, there are some variations in travel times, influenced by factors such as the proximity of destinations to residential areas or traffic conditions. On weekdays, shorter travel times might occur when schools or workplaces are closer to homes, while traffic congestion could extend travel durations. On weekends, travel times vary due to the flexible nature of leisure travel, but they generally represent shorter distances.

### 3.3. Mining Hotspots in Residents' Travel Space

### 3.3.1. Hotspot Area Identification

The spatial characteristics of residents' travel are as significant as the temporal ones, primarily represented through hotspot areas and paths. Identifying and analyzing these hotspots are crucial to understanding residents' travel behaviors.

Given that residents' travel activities on weekdays are more flexible and less geographically bound, making it challenging to detect travel hotspots, the analysis focused on weekdays. Representative time periods for this analysis were 7:00–9:00, 11:00–13:00, and 19:00–21:00. The OD point data for these intervals were extracted and subjected to the HDBSCAN algorithm for clustering. This process identified areas with a high concentration of pick-up and drop-off points. Subsequently, kernel density analysis on these high-density OD points was performed to visualize the hotspots of residents' travel.

(1)     The heat map for the 7:00–9:00 time period, as shown in Figure 13.

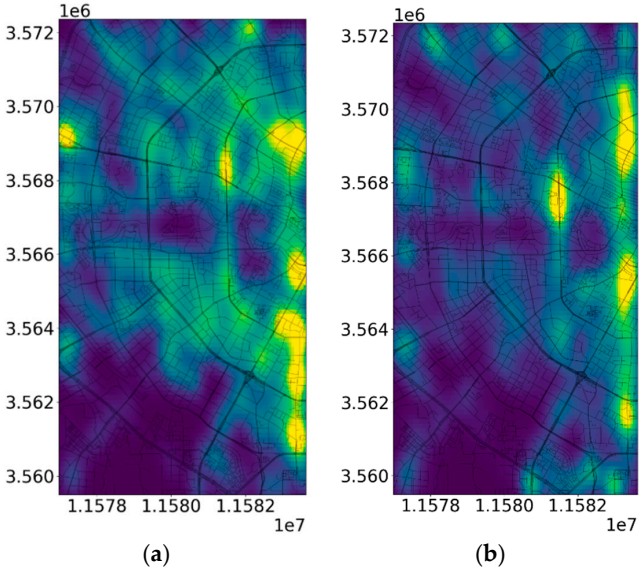

**Figure 13.** Heat map of pick-up and drop-off points from 7:00 to 9:00: (**a**) heat map of pick-up points; (**b**) heat map of drop-off points.

Figure 13 reveals that pick-up points are predominantly located in residential areas, transportation hubs, and commercial hotels, such as Yipintianxia Community, Tianfu Plaza, and Fengqiao Xiaoyue Elevator Apartment. In contrast, drop-off points are more common in areas associated with work, like schools, businesses, and government institutions, including Chengdu Jinxi Middle School, Mingyang Building, and Sichuan Orthopedic Hospital. When comparing the two heat maps, we can observe that during the morning peak, pick-up points are somewhat dispersed, resulting in more prominent hotspots for boarding. In contrast, drop-off points exhibit a more concentrated distribution, with fewer and more tightly concentrated hotspots. This disparity can primarily be attributed to the scattered nature of residential areas and transportation hubs, whereas schools, businesses, and government institutions tend to be clustered. This distribution pattern of hotspots aligns with the travel patterns of residents.

(2)     The heat map for the 11:00–13:00 time period, presented in Figure 14.

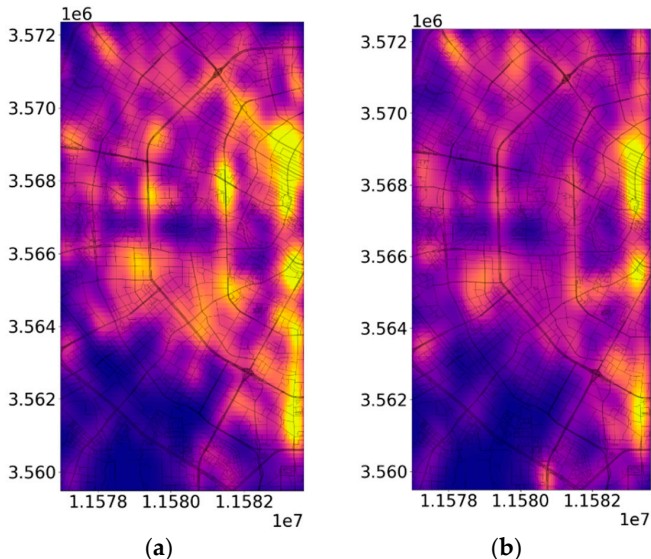

**Figure 14.** Heat map of pick-up and drop-off points from 11:00 to 13:00: (**a**) heat map of pick-up points; (**b**) heat map of drop-off points.

Figure 14 illustrates numerous hotspots for both taxi pick-up and drop-off points, each notable in its magnitude. Pick-up points are mainly clustered around schools, business centers, government agencies, and transportation hubs. An example is the vicinity of West Street, adjacent to Chengdu Third People's Hospital, Chengdu Shude Experimental Middle School, and Qingyang District People's Government. The presence of these medical, educational, and governmental institutions contributes to the large and intensely colored hotspots in the West Street area. The distribution of drop-off points is similar, though slightly fewer in number compared with pick-up points. Considering residents' tendency to spend short periods near their workplaces or schools during weekday peak hours, the distribution of pick-up and drop-off points in these core areas is relatively similar.

(3)     During the evening peak hours of 19:00–21:00, the heat map, as illustrated in Figure 15.

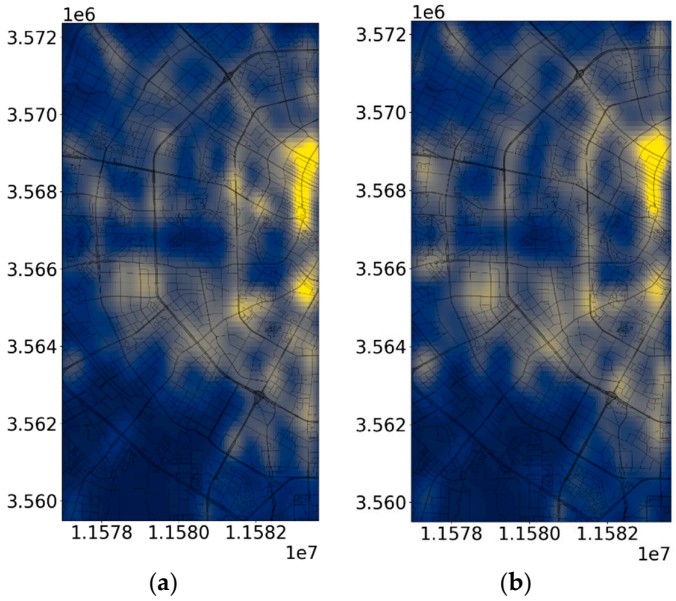

**Figure 15.** Heat map of pick-up and drop-off points during 19:00–21:00: (**a**) heat map of pick-up points; (**b**) heat map of drop-off points.

Figure 15 displays a widespread distribution of taxi pick-up and drop-off points across various points of interest like entertainment venues, dining areas, residential areas, offices, and business centers. This period witnesses significant taxi demand, with residents enjoying a high degree of travel freedom. Although many hotspots are formed, only a few exhibit exceptionally high density. Examples include business centers like Zhongkai Building and Jinke Shuangnan Tiandu, government agencies like the Sichuan Provincial Water Conservancy Planning Institute, and schools such as Wuhou Experimental Primary School. Drop-off points are frequently entertainment venues (Wanda Plaza, Jinli Ancient Street), dining areas (Longsenyuan Hotpot), and residential communities (Chengdu Garden). This phenomenon occurs because, during this time period, the majority of residents have typically completed their work or school commitments and have the option to participate in leisure or dining activities or return to their residential areas independently. The distribution pattern on the heat map also corresponds to the evening travel characteristics of residents.

3.3.2. Hotspot Path Mining

Hotspots are intricately connected to hotspot paths. Hotspot areas serve to pinpoint regions with high flow and density of residents' travel activities, while hotspot paths offer a detailed description of the paths characterized by high flow and density in residents' travel. The synergy of these two aspects enables a comprehensive exploration of residents' travel hotspots. Notably, hotspot path analysis was conducted on the foundation of hotspot areas, and the primary steps involved are as follows:

(1) Road network matching: Trajectory data points are spatially matched to the nearest road network segment using the hidden Markov model method. This step associates each trajectory point with road segments, providing related information like road name and vehicle passenger status.

(2) Hotspot detection: The geographical coordinates in the trajectory data are mapped to a discrete spatial grid for space discretization. Kernel density analysis is then applied to calculate the density of pick-up and drop-off points in each grid, identifying high-density areas and streets.

(3) Road extraction: This involves determining the geographical coordinates of the start and end points of hotspot paths, followed by path searching using the Dijkstra algorithm based on passenger volume, and finally recording and exporting the generated paths.

(4) Chi-square test: A crosstab is created based on hotspot areas and paths, and chi-square statistics and $p$-values are calculated under the null hypothesis. A $p$-value below 0.05 suggests a significant relationship between hotspot areas and paths.

Based on the analysis of hotspot paths during the specified time periods (7:00–9:00, 11:00–13:00, and 19:00–21:00), the visualization and analysis yield the following insights:

(1) During the 7:00–9:00 time period, as shown in Figure 16.

The hotspot paths for pick-up and drop-off points are primarily located within the identified hotspot areas, with their distribution closely mirroring these hotspots. Key pick-up locations include streets like Dongyu Street (near Tianfu Plaza and Chengdu Garden Hotel) and Jinli Middle Road (near Jinjiang Apartment and Jinjiang Times Garden). These areas are rich in residential buildings, hotels, and key transportation paths. Popular drop-off spots are located on streets such as Xinguanghua Street (near Sichuan Provincial Department of Education and Shishi United Middle School), Jiaodong Road (near Middle School Affiliated to Southwest Jiaotong University and Jinniu Civic Center), and Furong Avenue (near Wenjiang Campus of West China Hospital and Chengdu Food Inspection Institute). They are all proximate to educational institutions, government offices, and medical facilities. In general, during the 7:00–9:00 time period, the hotspots for pick-up tend to be associated with residential areas and transportation hubs, while the hotspots for drop-off are linked to schools, commercial buildings, and government offices. This

distribution of hotspots during this time period closely aligns with the peak travel patterns of residents.

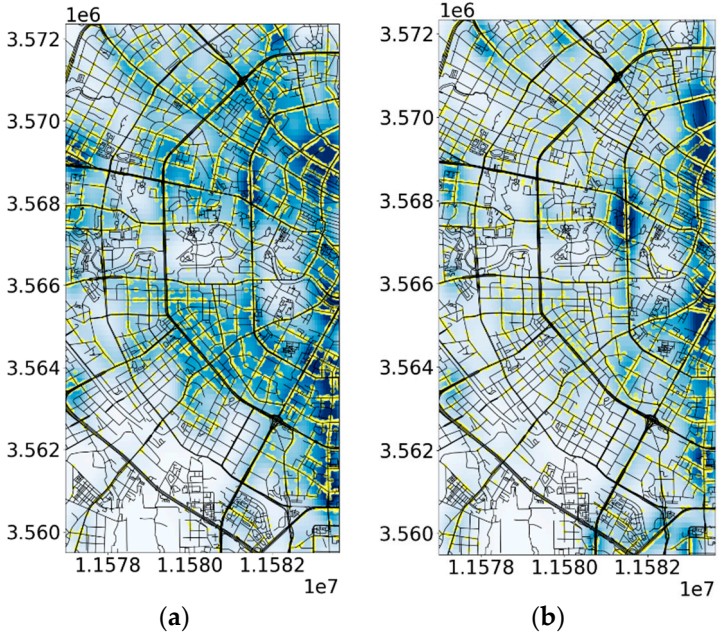

**Figure 16.** Hotspot path map of pick-up and drop-off points during 7:00–9:00: (**a**) hotspot path for pick-up points; (**b**) hotspot path for drop-off points.

(2) The visualization of hotspots for pick-up and drop-off points during the 11:00–13:00 time period is displayed in Figure 17.

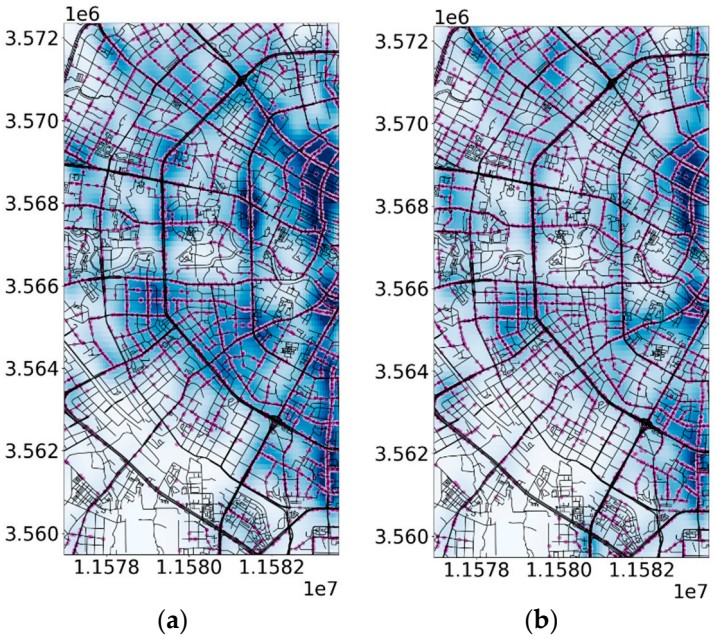

**Figure 17.** Hotspot path map of pick-up and drop-off points during 11:00–13:00: (**a**) hotspot path for pick-up points; (**b**) hotspot path for drop-off points.

The hotspots for pick-up during this time frame are primarily concentrated along streets such as West Street (near Chengdu Third People's Hospital and Second Middle School Hospital), Shuhan Street (near the People's Government of Wuhou District and Wuhou Campus of Southwest University for Nationalities), and Jinyang Road (near Longhu

Chengdu Jinnantian Street and Jinyan Building). Meanwhile, hotspots for drop-off are mainly found in areas like Ningxia Street (near New City Plaza and Golden Hawaii), Gaoshengqiao Road (near Jinli Ancient Street and Jincheng Impression Hotpot Restaurant), and Changshun Shangjie (near Kuanzhai Alley and Zuncheng International). Analyzing these hotspot paths in conjunction with the specific functions of these streets reveals that pick-up points are concentrated near commercial buildings, campuses, government offices, and so forth, while drop-off points are concentrated around restaurants, shopping streets, and other leisure and entertainment venues. This distribution aligns with the travel patterns of residents during the weekday afternoon peak period.

(3) In Figure 18, we can observe the visualization of hotspots for pick-up and drop-off points during the 19:00–21:00 time period.

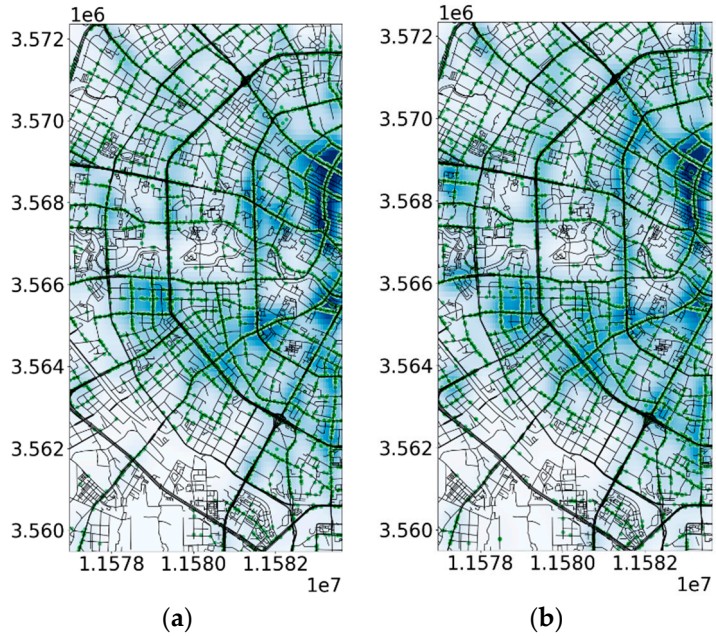

(**a**)  (**b**)

**Figure 18.** Hotspot path map of pick-up and drop-off points during 19:00–21:00: (**a**) hotspot path for pick-up points; (**b**) hotspot route for drop-off points.

Notably, the distribution of hotspots during this time frame is more extensive compared with other time periods. The volume of travel is relatively evenly spread, with only a few specific hotspot paths carrying a significant number of passengers. Hotspots for drop-off are primarily located in areas like Luoguoxiang (near New Times Square and Chengdu Municipal People's Government Government Service Center), West First Ring Road (near Wuhou Campus of Chengdu Institute of Physical Education and Gaosheng Building), and Jiangxi Street (near Sichuan Provincial Committee of the Chinese People's Political Consultative Conference and Hongda International Plaza). On the other hand, drop-off hotspots are mainly found at Fangcao West Second Street (near Jinxingyuan and Golden Age), the Qingyang Avenue section of Middle Ring Road (near Xiaotaoyuan Restaurant and Chengdu Garden Plaza), and Shuangxin South Road (near Cuiliu Garden and Twenty First Century Garden). A comparison reveals that during the evening peak period, pick-up and drop-off points tend to shift away from workplaces and school districts toward dining venues, entertainment spots, and residential areas. This is because residents have more free travel time and flexibility during this part of the day, allowing them to make travel choices that align with their leisure and lifestyle preferences. This observation confirms the earlier analysis of the widespread distribution of hotspot areas during this particular time period in Section 3.3.1.

(4) Information regarding the geographical boundaries of hotspot areas and the paths commonly taken for pick-up and drop-off activities during the time intervals of

7:00–9:00, 11:00–13:00, and 19:00–21:00 was gathered. A cross tabulation table that combines hotspot areas and paths for residents' travel was created, as presented in Table 3.

**Table 3.** Cross tabulation table example.

| Hotspot | Tianfu Square | | | Yindu Building | | | Taurus Park | | |
|---|---|---|---|---|---|---|---|---|---|
| Time Period Hot Path | M | N | E | M | N | E | M | N | E |
| Shudu Avenue | 982 | 683 | 1249 | 0 | 0 | 0 | 0 | 0 | 0 |
| Xiyu Street | 844 | 631 | 986 | 0 | 0 | 0 | 0 | 0 | 0 |
| Xinhua Avenue | 0 | 0 | 0 | 893 | 702 | 1093 | 0 | 0 | 0 |
| Renmin Middle Road | 0 | 0 | 0 | 947 | 816 | 1276 | 0 | 0 | 0 |
| Jinfu Road | 0 | 0 | 0 | 0 | 0 | 0 | 1216 | 935 | 1427 |
| Seedling Road | 0 | 0 | 0 | 0 | 0 | 0 | 893 | 537 | 1046 |

Table 3 displays a subset of the cross tabulation data involving hotspot areas and paths. Given the extensive volume of data, importing it into Python for analysis is a practical approach. The calculated $p$-value for the chi-square test concerning hotspot areas and paths is 0.023. When comparing this $p$-value with the significance level (typically set at 0.05), it is evident that both values are less than 0.05. This suggests a significant correlation between hotspot areas and hotspot paths.

## 4. Related Work

(1)　Use the clustering algorithm to mine hotspot areas.

Wang et al. employed taxi trajectory data points and utilized kernel density clustering analysis to group the trajectory data into hotspots. They further categorized these clustering areas into different levels of hotspots and generated spatial distribution maps representing these hotspots [29]. In a related study, Zhao et al. enhanced the traditional K-means algorithm by proposing a clustering model that incorporates self-organizing mapping and K-means techniques. This modification reduced the computation time required for the hotspot clustering model and ultimately yielded stable travel hotspot regions [30]. Such research endeavors aim to provide consistent clustering of residents' travel locations. However, it is worth noting that the boundaries of clustered trajectory points may become less distinct when dealing with varying amounts of data.

(2)　Improve the clustering algorithm to improve the clustering accuracy of trajectory data.

Wu et al. introduced fundamental time threshold control parameters into the density clustering approach. They segmented the data based on time granularity and performed trajectory clustering according to specific time periods [31]. Meanwhile, Zhou et al. presented a framework centered on division and grouping for trajectory clustering. This approach enhanced the precision of local trajectory point clustering within a specific geographical range [32]. These research efforts enable more precise control and analysis of clustering in both time and space dimensions. However, it is important to note that the clustering results can still be influenced by the configuration of clustering parameters.

(3)　Analyze residents' travel mechanisms by mining travel hotspots

Wang et al. employed Markov decision-making techniques to learn from hourly taxi data. They conducted relevant parameter calculations to determine the optimal strategies for taxi movements that would maximize revenue [33]. In a related study, Ma et al. took into account various factors, including cost and distance, to identify recommended peak travel times. They suggested that taxi drivers should focus on searching for passengers during these peak hours [34]. Sun et al. conducted statistical analysis using taxi trajectory data in conjunction with precipitation data. They found that on rainy days, during the morning peak period, the demand for taxis significantly exceeded the available supply [35].

Similarly, Oleyaei-Motlagh and Vela analyzed the relationship between weather conditions and taxi travel demand. They discovered that rain and snow weather led to an increase in taxi travel demand, mainly driven by residents opting for shorter-distance trips [36]. This kind of research provides valuable insights into the primary travel patterns and dynamic changes in a city's residents. It aids in the development of more effective transportation planning and taxi management policies. However, it is important to acknowledge that research results can be influenced by the natural environment and human factors affecting the collection of trajectory data.

(4) Solve the unbalanced supply and demand of transportation resources through travel characteristics.

Li et al. predicted residents' activities by analyzing the number of passengers in urban hotspot areas. They also proposed an enhanced prediction method based on the autoregressive integrated moving average (ARIMA) model to forecast spatiotemporal changes in passenger volumes within hotspot areas. This information was used to provide taxi drivers with recommendations for passenger hotspot areas [37]. Zheng et al. simulated the dynamic customer-seeking behavior of taxi drivers under uncertain conditions. They associated dynamic demand planning with the cumulative distribution function of user waiting time, leading to the development of a dynamic taxi demand prediction model [38]. Abolfazl et al. introduced a generalized spatiotemporal autoregressive model and employed a LASSO-type penalty method to address the model's high dimensionality. This approach improved the prediction efficiency of the model to a certain extent [39]. Technicolor et al. proposed an end-to-end deep learning model, incorporating a deep belief network to analyze nonlinear dependencies, to address the travel demand prediction challenge [40]. Xu et al. conducted statistical analysis and data mining on taxi trajectory data in a California study area. They identified characteristic travel time patterns of residents in the region and predicted the upcoming peak tourist season [41]. This type of research leverages trajectory data to capture and analyze trends in time series data, aiding in the prediction of spatiotemporal changes in passenger volumes. However, it is important to note that prediction models heavily rely on a large volume of experimental data, and their accuracy can be affected by natural weather conditions and man-made policies.

## 5. Conclusions

This research was built upon extensive taxi trajectory data and employed various analytical methods, including mathematical statistics, the HDBSCAN clustering algorithm, kernel density analysis, and hidden Markov model road matching. Starting with an examination of both time and space, the research scrutinized residents' behavior across varying time periods and geographical regions. This analysis included travel patterns and extracted key hotspot areas and paths preferred by residents within the study area. Additionally, it confirmed the connections between these hotspots, with the primary findings summarized as follows:

(1) Mathematical statistical analyses, individual assessments, and comparative examinations of taxi pick-up and drop-off point traffic volumes during distinct time intervals on weekdays and weekends were conducted. The investigation reveals that on weekdays, residents' travel times range from 19 to 40 min, while on weekends, the average travel time falls within the range of 16 to 35 min. Furthermore, it is observed that the morning and afternoon peak hours for residents' travel on weekdays occur earlier compared with those on weekends.

(2) A fusion algorithm was used to cluster OD points on both weekdays and weekends, resulting in spatially continuous clusters. Kernel density analysis provided an accurate heat map of pick-up and drop-off points after eliminating the noise influence, identifying travel hotspots. Different functional area travel characteristics were observed: During the morning peak (7:00–9:00), taxis are frequently used in residential areas and transportation hubs, with drop-offs common at schools, commercial build-

ings, and government agencies. In the afternoon peak (11:00–13:00), travel shifts from schools, businesses, and government locations to dining and entertainment areas. In the evening peak (19:00–21:00), pick-up and drop-off points are widely distributed, moving from schools, commercial zones, and residential areas to dining streets, shopping plazas, and entertainment venues.

(3) Utilizing the hidden Markov model approach, trajectory points were matched with the road network. Hotspot detection was subsequently conducted based on these hotspot areas, resulting in the identification of hotspot paths. Through the application of the chi-square test method, it was determined that the *p*-value from the chi-square test is 0.023, which is less than the conventional significance threshold of 0.05. This statistical analysis takes into account spatial changes in hotspot areas and the count of hotspot paths across different time periods. The result provides strong evidence to support the assertion that hotspot areas and hotspot paths exhibit a significant relationship.

Future studies aim to utilize low-frequency taxi trajectory data to develop clustering methods with enhanced levels and accuracy. By integrating urban points of interest (POI) data, the research will further refine the identification of hotspot areas and paths for residents' travel. This will deepen the understanding of the correlation between hotspot areas and paths, offering more effective decision support for urban transportation planning and management.

**Author Contributions:** Conceptualization, J.D., C.M. and X.L.; methodology, J.D.; software, C.M.; validation, J.D. and C.M.; formal analysis, J.D. and X.L.; investigation, C.M. and X.L.; resources, J.D.; data curation, J.D.; writing—original draft preparation, C.M.; writing—review and editing, J.D. and X.L.; visualization, J.D.; supervision, J.D.; project administration, J.D.; funding acquisition, J.D. All authors have read and agreed to the published version of the manuscript.

**Funding:** This research was funded by the National Natural Science Foundation of China (grant number 41801318) and the Key Scientific and Technological Project of Henan Province (grant numbers 232102321004 and 212102310436).

**Institutional Review Board Statement:** Not applicable.

**Informed Consent Statement:** Not applicable.

**Data Availability Statement:** The data presented in this study are available on request from the corresponding author.

**Acknowledgments:** We sincerely thank all the transportation departments and taxi drivers in Chengdu who participated in this study and supported the data collection. We also thank the owners and organizers of the Data Castle data research platform for providing access to the data collection.

**Conflicts of Interest:** The authors declare no conflicts of interest.

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
