# Peer review of "Analysis of Urban Residents’ Travelling Characteristics and Hotspots Based on Taxi Trajectory Data"

_applsci, doi:10.3390/app14031279_

Round 1

Reviewer 1 Report

Comments and Suggestions for Authors

file is attached

Comments on the Quality of English Language

Author Response

Reviewer #1

  1. The work is commendable in its entirety, and its readiness for publication is apparent, contingent upon certain refinements. Specifically, it warrants an update of its literature review to encompass the most recent publications in the field regarding winter traffic patterns based on classified vehicle types. Moreover, the manuscript could benefit from the inclusion of forward-looking themes and advanced topics for potential exploration in future research.

Reply: Thank you for your insightful suggestion. In response, we have expanded our literature review to include recent studies on resident travel patterns using vehicle trajectory data (refer to updated references [4]-[16]). This addition enriches our analysis and sets a foundation for future research directions.

  1. The readability of Figures 4 to 7 is impeded due to issues with the legends, as well as the titles of the x and y-axes. It is recommended that these elements be revised for improved clarity and comprehension.

Reply: We appreciate your feedback on this aspect. We have enhanced the legibility of these figures by adjusting the font size of the legends and revising the axis titles for clarity. The updated Figures 4-7 are now included in the manuscript for your review.

Reviewer 2 Report

Comments and Suggestions for Authors

The paper is well structured, and clearly focuses on its goal.

Although the study is interesting and worthy of further study, it would be appropriate to estimate the number of trips made by taxi on the total volume of trips by residents (#totaltrips/#residentstrips), otherwise the analysis could lead to errors in evaluation (e.g. if the percentage was low, the result of the survey could not be referable to the entire class of residents' users).

Furthermore, regarding section 2.1 "Preprocessing of trajectory data", what criterion is behind the choice of the thresholds of 6 ms and 16.7 ms? Have any analyzes been carried out on the acquired data sample?

Author Response

Reviewer #2

  1. Although the study is interesting and worthy of further study, it would be appropriate to estimate the number of trips made by taxi on the total volume of trips by residents (#totaltrips/#residentstrips), otherwise the analysis could lead to errors in evaluation (e.g. if the percentage was low, the result of the survey could not be referable to the entire class of residents' users).

Reply: Thank you for highlighting this important aspect. The passenger capacity of taxi in section 3.2 represents the overall number of taxi trips taken by residents. By sorting and analyzing the OD points extracted in section 2.2 based on time, and tallying the boarding and alighting frequencies, we can derive the total number of resident trips, which is equivalent to the number of taxi passenger trips, sorting by carb-id gives us the total number of taxis in operation. Consistent with your recommendations.

  1. Furthermore, regarding section 2.1 "Preprocessing of trajectory data", what criterion is behind the choice of the thresholds of 6 m/s and 16.7 m/s ? Have any analyzes been carried out on the acquired data sample?

Reply: We are grateful for your request for clarification. The selection of these speed thresholds was determined based on varying road speed limits within the study area. We have provided a detailed explanation, along with an analysis of the taxi trajectory data in relation to these limits, in the revised section 2.1 (highlighted in yellow for ease of reference).

Reviewer 3 Report

Comments and Suggestions for Authors

Refer to the attached review report. 

Author Response

Reviewer #3

  1. However, the provided literature review study seems insufficient. Since the study involved

various analytical methods, the selection of each method and technique should be justified through a comprehensive literature review. Perhaps, the related work section should be relocated to the early section of the paper and become a part of the literature review. In addition, the presentation of the analysis findings in Section 3.3 could be improved by putting the figure before the paragraph that discusses that figure. For instance, Figure 14 should appear before the paragraph in the Line 455. It’s easier for the reader to observe the graph before reading the elaboration of the graph.

Reply: Thank you for your suggestions. We have expanded the literature review, adding comprehensive information about clustering algorithms like DBSCAN and HDBSCAN, and various assessment models used for analyzing residential travel hotspots (see updated references [4]-[16]).

  1. Moreover, some mathematical notations require minor improvements for better presentation.

Other than that, some of the provided figures are insufficiently clear. Sometimes, quite difficult to read characters and numbers on the figures. Since most of the results and insights are interpreted from the figures, better quality figures are highly recommended.

Some corrections are suggested below for the betterment of this paper:

Title: The term “urban” should be included in the title.

Equation (1): ????(?????, ??????) → ????(?????, ??????)

Line 186 & Equation (4): ??? → ??,? – put a comma between ? and ?

Equation (4): What is the input for the first cos function in this equation?

Equation (4): ???1, ???2 → ???1, ???2

Equation (4): Parameter ? is suddenly appear without being defined or explained.

Figures 4, 6, 7, 8, 9, 10, 11, 13, 14, 15, 16, 17, 18: Blur and difficult to read.

Line 263: ?? → ??

Figure 9, 10, 11: Captions and figures are separated to the next page. Relocate the captions

to make them unseparated with the corresponding figures respectively.

Line 376 – 390: The authors interchangeably used 12hour and 24 hours system. To avoid

confusion to the reader, standardise to 24 hours system in the whole paper.

Line 395: ?1, ?2, ?3, ?4,… ?? → ?1, ?2, ?3, ?4, … ?? & ∑ ??? = 1 →

Reply: We appreciate your attention to detail. In response to your feedback, we have corrected and clarified the mathematical notations in Equations (1) and (4). The quality of Figures 4, 6, 7, 8, 9, 10, 11, 13, 14, 15, 16, 17, and 18 has been improved for better legibility, ensuring that the captions remain with the corresponding figures. We have standardized the time presentation to a 24-hour system in Lines 425-433. These changes are highlighted in yellow in the manuscript for easy reference.

  1. The term “urban” should be included in the title.

Reply: Thank you for this insightful suggestion. We have revised the title to "Analysis of Urban Residents' Traveling Characteristics and Hotspots Based on Taxi Trajectory Data", which more accurately reflects the scope and focus of our study.

We trust that these revisions adequately address your concerns and contribute to the improvement of our manuscript. Should you have any additional suggestions or require further clarifications, we are more than willing to make the necessary adjustments.

Thank you once again for your invaluable feedback and the time dedicated to reviewing our work. We eagerly await your decision on our revised manuscript and remain committed to enhancing its quality and relevance to the field.